# Optimal Intermittent Administration Interval of Abaloparatide for Bone Morphogenetic Protein-Induced Bone Formation in a Rat Spinal Fusion Model

**DOI:** 10.3390/ijms25073655

**Published:** 2024-03-25

**Authors:** Tetsutaro Abe, Masashi Miyazaki, Noriaki Sako, Shozo Kanezaki, Yuta Tsubouchi, Nobuhiro Kaku

**Affiliations:** 1Department of Orthopaedic Surgery, Faculty of Medicine, Oita University, Oita 879-5593, Japan; abe-te@oita-u.ac.jp (T.A.); nobuhiro@oita-u.ac.jp (N.K.); 2School of Physical Therapy, Faculty of Rehabilitation, Reiwa Health Sciences University, Fukuoka 811-0213, Japan

**Keywords:** abaloparatide, bone morphogenetic protein, bone fusion, rat spinal fusion model

## Abstract

Both bone morphogenetic protein 2 (BMP-2) and abaloparatide are used to promote bone formation. However, there is no consensus about their optimal administration. We investigated the optimal administration theory for the pairing of BMP-2 and abaloparatide in a rat spinal fusion model. Group I was only implanted in carriers and saline. Carriers with 3 µg of recombinant human BMP-2 (rhBMP-2) were implanted in other groups. Abaloparatide injections were administered three times a week for group III (for a total amount of 120 µg/kg in a week) and six times a week for group IV (for a total amount of 120 µg/kg in a week) after surgery. They were euthanized 8 weeks after the surgery, and we explanted their spines at that time. We assessed them using manual palpation tests, radiography, high-resolution micro-computed tomography (micro-CT), and histological analysis. We also analyzed serum bone metabolism markers. The fusion rate in Groups III and IV was higher than in Group I, referring to the manual palpation tests. Groups III and IV recorded greater radiographic scores than those in Groups I and II, too. Micro-CT analysis showed that Tbs. Sp in Groups III and IV was significantly lower than in Group I. Tb. N in Group IV was significantly higher than in Group I. Serum marker analysis showed that bone formation markers were higher in Groups III and IV than in Group I. On the other hand, bone resorption markers were lower in Group IV than in Group I. A histological analysis showed enhanced trabecular bone osteogenesis in Group IV. Frequent administration of abaloparatide may be suitable for the thickening of trabecular bone structure and the enhancement of osteogenesis in a rat spinal fusion model using BMP-2 in insufficient doses.

## 1. Introduction

There are few reports on accelerated bone fusion in spinal fusion models using various drug combinations. Spinal fusion is the radical treatment of spinal disorders and one of the most general spinal procedures, with over 200,000 procedures annually in the USA [1]. Bone grafting is used to form a bony fusion between the vertebrae and restore the stability of the spine. The completion of osteosynthesis between vertebrae with instability leads to the relief of pain and convalescence from neurological symptoms, and its effectiveness is widely recognized, with the number of such surgeries increasing every year as the elderly population grows [2,3,4,5].

Bone morphogenetic proteins (BMPs) are transforming growth factor-β superfamily members and are potent bone-inducing molecules [6]. BMPs are also thought to promote osteoclast genesis, as they are receptor activators of the nuclear factor kappa beta ligand, stimulate osteoblast production, help mature osteoclasts survive, and may be involved in bone resorption [7,8]. Recombinant human BMP-2 (rhBMP-2) for spine fusion has been shown to have osteoinductive effects in animal models and clinical studies [9,10,11,12,13]. Though rhBMP-2 is approved for clinical use in the USA, clinical trials have shown higher doses are required to induce sufficient bone fusion for the following reasons: (1) the solubility of the molecule, (2) the ease of diffusion of the molecule from the fusion site, and (3) inactivation in vivo [14]. Furthermore, BMPs are expensive, and their utility may be limited. Consequently, several strategies have been developed for safer, cheaper, and more effective spine fusion techniques by using rhBMP-2.

The parathyroid hormone-related protein (PTHrP) is at least equivalent to PTH in the first 15 residues, sterically equivalent to the 34 residues of PTH, and expresses its bone action via the PTH receptor 1 (PTHR1). Abaloparatide, a PTHrP derivative, matches the first 22 residues of PTHrP, but its receptor-mediated action on the bone is very different from that of PTHrP. PTHR1 has different forms of receptors, called RG and R0, and binding to RG provides only a short-lived PTH signal and promotes osteogenesis [15,16,17,18]. Binding to R0 sustains PTH signaling and promotes bone catabolism. This indicates that the strength of the bone anabolic effect depends on the ratio of RG-mediated to R0-mediated signaling. Both abaloparatide and teriparatide have higher affinities for RG than R0; however, teriparatide has a 3-fold higher affinity, whereas abaloparatide has a 1600-fold higher affinity [19].

We previously reported that more frequent teriparatide administration may thicken and strengthen the trabecular constituent of newly formed bone tissue in a rat spinal fusion model with inadequate BMP-2 [20]. The aim of the present study was to investigate the optimal dosing interval for the combination of low-dose rhBMP-2 and abaloparatide using a rat spinal fusion model.

## 2. Results

No abnormal behaviors or neurological symptoms were observed in all rat perioperative periods. No wound infection was observed in any of the rats. There were no significant differences in weight gain between the groups until euthanasia.

### 2.1. Manual Palpation

Table 1 shows the percentage of bone fusion achieved in each group during manual assessment. Consistent agreement (κ = 0.821) was found between the three independent observers.

Eighteen segments in Group III were assessed as fused (fusion rate 75.0%). Sixteen segments in Group IV (fusion rate 66.7%) and four segments in Group II (fusion rate 16.7%) were assessed similarly. In Group I, not a single specimen was assessed as fused. (fusion rate 0%). During manual assessment, the fusion rate was significantly higher in Groups III and IV compared to Groups I and II (*p* < 0.05). On the other hand, no significant differences were found between Groups III and IV.

### 2.2. Radiographic Analysis

Spinal radiographs were obtained after the rats were euthanized. There was consistent agreement (κ = 0.824) between the observers who assessed this analysis. Evaluations were carried out on both the left and right sides. The scores in each group are shown in Table 2. Representative images of the bone fusion status of each group are also shown in Figure 1. Osteogenesis and bone bridging were detected between the L4 and L5 transverse processes in Groups III and IV. Mineralized calli bridging the L4 and L5 transverse processes were detected in Group II but in insufficient quantities. Group I showed no evidence of osteogenesis. Groups III and IV had significantly greater radiographic scores than Groups I and II (*p* < 0.05).

### 2.3. Micro-CT Analysis

Micro-CT images showed the amount of newly formed bone and its quality. The average micro-CT data histomorphology of each group is indicated in Table 3 and Table 4. In Figure 2, the representative 3D images are shown. The data made significant differences clear in the percentages of bony mass. The BV scores were significantly greater in Groups Ⅲ and IV compared with Groups I and Ⅱ. The BV/TVs in Group IV were significantly greater than those in other Groups. Tb. N was greater in Group IV than in Group I. Tb. Sp in Groups III and IV was significantly smaller than that in Group I.

### 2.4. Serum Markers of Bone Metabolism

An enzyme-linked immunosorbent assay showed serum osteocalcin in Groups III and IV was significantly higher than in Group I (*p* < 0.05) (Figure 3). Similarly, serum TRACP5b levels in Group IV were significantly lower than in Group I (*p* < 0.05) (Figure 4).

### 2.5. Histological Analysis

The Group I histological analysis showed no evidence of fusion and no new bone formation (Figure 5A,B). Muscle fibers are clearly visible between the transverse processes and no bony tissue is visible. Minor evidence of new bone formation was observed, which may have resulted from the shedding of the transverse processes or from normal remodeling. In Group II, there was a distribution of cartilage tissue, but no evidence of bony fusion with necrotic tissues and muscular fibers (Figure 5C,D). Group III indicated cartilage tissue and immature bony formation, but no mature bony tissues in that area (Figure 5E,F). The neoplastic bony formation, mature bone tissue, and contracted bone beams were cross-linked between the transverse processes in Group IV (Figure 5G,H). Table 5 presents histological fusion scores. In Groups III and IV, the fusion scores were significantly greater than in Groups I and II (*p* < 0.05). Consistent agreement (κ = 0.821) was found between observers.

## 3. Discussion

This study provides preliminary evidence that abaloparatide, which is a selective PTH1R agonist with 76% homology to PTHrP, promotes vertebral bone healing in rats with insufficient rhBMP-2. Past research has shown that genetic PTHrP deficiency reduces cartilage formation and bone callus development in fracture conditions [21], while exogenous PTHrP and PTHrP analogs promote the healing of fractures and osteotomies [22,23]. A recent study in a rat posterolateral spinal fusion model also demonstrated that daily abaloparatide administration (20 µg/kg/day) significantly increased bone formation [24]. This study showed that frequent abaloparatide administration, even in combination with BMP-2, promoted bone formation, fracture bridging, and biomechanical recovery. In contrast, the TV had significant differences between Group III and Group IV. This may have been due to progressive remodeling with frequent abaloparatide administration. These findings suggest that high-frequency abaloparatide administration may improve the bone fusion rate in spinal fusion procedures and vertebral fractures.

Differences in the biological actions of abaloparatide and teriparatide might result in different mechanisms of action. Differences in the binding of abaloparatide compared to teriparatide result in conformational binding selectivity in favor of parathyroid type 1 receptor anabolism [19,25,26]. As a result, abaloparatide strongly stimulated cyclic adenosine monophosphate production in osteoblasts and reduced the osteoblast-derived RANK ligand expression. This may have resulted in reduced bone resorption.

Abaloparatide increased osteocalcin, which is a bone formation marker, compared to Group I. Serum osteocalcin has been reported to be associated with the fused bone mass microstructural parameters [24], further suggesting that the osteoblast-stimulating effects of abaloparatide can affect the bone fusion process. In the study of mice, both teriparatide and abaloparatide increased bone resorption markers; on the other hand, bone formation markers were significantly higher in abaloparatide than in teriparatide [27]. In this study, serum TRACP-5b was reduced in the six-times-weekly abaloparatide injection group compared with the control group, suggesting more frequent dosing may favor osteogenesis by expanding the anabolic window. The reason for the reduced TRACP5b in the six-times-weekly abaloparatide group is that it is expected to have less effect on promoting bone resorption compared to teriparatide, especially in short-term administration [15]. In long-term observations, this effect is expected to be limited.

The pharmacodynamic actions that promote bone formation with minimal impact on bone resorption are desirable in patients undergoing spinal fusion procedures. This is because insufficient assimilation or premature catabolism of fixed masses can lead to pseudo-articulation [28]. Some observations indirectly supported this hypothesis. For example, low bone formation marker levels and high serum TRACP-5b levels are important risk factors for non-fusion in spinal fusion surgery [29]. The prospective controlled clinical study showed an increased rate of early fusion in patients receiving weekly teriparatide, which increased bone formation markers with no ascending bone resorption markers [30,31]. In contrast, a recent placebo-controlled prospective study also showed that the daily administration of standard teriparatide, which increases bone formation and resorption markers, did not affect endpoints related to joint formation or the amount of fusion [32]. In another study, patients who received daily teriparatide, along with t denosumab, which is a potent osteoclast inhibitor, had a higher fusion rate at 6 months compared to those who received teriparatide alone [33]. Data from animal studies also showed that bone formation was promoted when the osteoclast-stimulating effects of BMP-2 were suppressed [34]. In this study, pathological evaluation showed that high-frequency abaloparatide administration, when combined with BMP-2, promoted better bone union by stimulating osteoblasts and inhibiting osteoclasts.

A limitation of this study is that the process of bone fusion could not be investigated. Only rats sacrificed at eight weeks postoperatively were identified. Micro-CT imaging requires the rats to remain still for approximately 30 min, which necessitates sacrificing the rats and removing their vertebrae before examination. Therefore, it is difficult to continue observing a living spine. Depending on the conditions under which abaloparatide is administered, fusion may occur more quickly; however, this study did not show complete fusion at 8 weeks. In addition, different methods of abaloparatide administration differentially affect bone metabolism markers, even at the same dose. Another limitation is that our results cannot be applied directly to humans because of the different biological responses of rats. Finally, although the total doses were aligned in this study, it remains debatable whether more frequent dosing is better, or whether the per-dose dosing was more appropriate in bone formation. Bone formation was more enhanced in Group IV, in which the frequency of dosing was increased, than in Group Ⅲ, in which single doses were increased. So, the frequency of administration was determined to have a greater impact in this study. Additional studies under various conditions are needed in the future. However, we believe that the information obtained in this study is useful for understanding the effects of these treatments.

The frequent systemic administration of abaloparatide had a positive influence on new bone formation by reducing bone resorption markers and increasing bone formation markers with inadequate BMP-2 levels in a rat spinal fusion model. Additional research in larger animals is required to better understand the efficacy of abaloparatide in spinal fusion procedures.

## 4. Materials and Methods

### 4.1. Preparation of Matrices

MedGEL^®^ (MedGEL, Kyoto, Japan) is a biodegradable gelatine hydrogel scaffold for cell adhesion. The final rhBMP-2 concentration (Peprotech, Rocky Hill, NJ, USA) was dissolved in phosphate-buffered saline (pH 7.5) and applied to MedGEL^®^. MedGEL^®^ was cut into 5 mm × 20 mm shapes and placed with rhBMP-2 in Eppendorf tubes. The tubes were left overnight at 4 °C before transplantation. And to obtain rhBMP-2-free MedGEL^®^, 100 µL of rhBMP-2-free phosphate-buffered saline was added to the MedGEL^®^.

### 4.2. Animals

All animal experiments were approved by the Oita University Animal Experimentation Committee and complied with all regulations and guidelines regarding animal welfare protection (protocol No. 1624002).

### 4.3. Study Groups

We divided 38 male Sprague–Dawley rats (8–10 weeks old; CLEA Japan, Inc., Tokyo, Japan) into four groups. Group I (*n* = 6) included rats with rhBMP-2-free MedGEL^®^. Group II (*n* = 12) included rats with carriers containing 3 µg of BMP-2, but without the administration of abaloparatide (Ostabalo; Teijin Pharma, Tokyo, Japan). Group III (*n* = 12) included rats with carriers containing 3 μg of BMP-2, and injections of abaloparatide (40 μg/kg) thrice a week (120 μg/kg/week). Group IV (*n* = 12) included rats with a carrier containing 3 μg of BMP-2, and injections of abaloparatide (20 μg/kg) six times a week (120 μg/kg/week). The rats in Groups III and IV were started on subcutaneous injections of abaloparatide one week after operation and continued until just before they were euthanized.

### 4.4. Surgical Technique of Rat L4–L5 Posterolateral Spinal Fusion Model

We made a posterior midline incision in the rat skin. The transverse processes were exposed through incisions on both sides, 3 mm lateral to the midline of the lumbar fascia. We decorticated the transverse processes of L4 and L5 with low-speed burrs. Then, MedGEL^®^ with or without BMP-2 was implanted on each side of the rat. The fasciotomy and skin incisions were closed using 3–0 absorbable sutures. The rats were administered analgesics (subcutaneous buprenorphine and paracetamol) after surgery and for the subsequent three days. Rats were placed in separate cages and fed ad libitum with food and water. We humanely euthanized the rats at 8 weeks postoperatively.

### 4.5. Manual Assessment of Fusion

Eight weeks after operation, the rat spine was removed and the fusion between the vertebrae was manually assessed. Three blinded independent observers carried out the assessments [35,36,37,38,39,40,41]. The L4–L5 levels were manually side-bent and compared with movements of the adjacent upper (L3–L4) and lower (L5–L6) levels. If there is no movement compared to the levels above and below, the fusion is considered successful. On the other hand, if movement is observed between the transverse processes, the fusion is considered to have failed. Fusion was only considered to have occurred if the assessments of all three observers were unanimous. Fusion and non-fusion of the spine were assessed on each side and the fusion rate was calculated.

### 4.6. Radiographic Analysis

The specimens were manually assessed for fusion and then filmed by a Softex X-ray system (Softex CSM-2; Softex, Tokyo, Japan) with HS Fuji Softex film (Fuji Film, Tokyo, Japan) at 45 cm, 30 kV, 15 mA for 20 s. The percentage of newly formed bone area between transverse processes was assessed [35]. Three blinded independent observers gave a rating on a 5-point scale from 0 to 4: 0 = without osteogenesis, 1 = bone formation in 25% or less of the area, 2 = bone formation in 25–50%, 3 = bone formation in 50–75%, 4 = bone formation in 75–100% between the transverse processes. The evaluation was carried out at two locations per sample, on the left and right sides, respectively.

### 4.7. Micro-CT Analysis

Eight weeks after surgery, the spine of a rat removed was micro-CT scanned with a voxel size of 20 µm using a SkyScan 1172 (Bruker micro-CT, Contich, Belgium). Reconstruction was performed using data collected with 100 mA and 100 kV. A cone beam algorithm was used for reconstruction. Each sample was set on the stage and scanned while being rotated 180°. The exposure time was 105 ms. The cylindrical volume of interest with a diameter of 20 mm and height of 27 mm was selected to represent the microstructure of vertebrae. The area from the upper end of the L4 transverse process to the lower end of the L5 transverse process was analyzed. Data analysis was performed using CT Analyzer software (Bruker micro-CT). The spine was analyzed in two locations, left and right, respectively. Tissue volume (TV), bone volume (BV), bone volume fraction (BV/TV), and trabecular thickness (Tb. Th), number (Tb. N), and spacing (Tb. Sp) were measured at each analysis site of the spine.

### 4.8. Analysis of Serum Markers

Blood samples were also collected immediately before the rats were euthanized. The samples were stored at −80 °C until analysis. Serum bone metabolism markers were identified using the enzyme-linked immunosorbent assay (RatTRAP Assay) specific for osteocalcin (Osteocalcin Highly Sensitive EIA Kit [rat]; Takara Bio, Shiga, Japan), which reflects bone formation. Tartrate-resistant acid phosphatase-5b (TRACP5b), which reflects bone resorption, and a specific enzyme-linked immunosorbent assay (RatTRAP Assay; Immunodiagnostic Systems Ltd., Boldon, UK) were used, too.

### 4.9. Histological Analysis

Specimens removed from rats were fixed in 40% ethanol and demineralized with 10% demineralizing solution HCI (Cal-Ex; Fischer Scientific, Fairlawn, NJ, USA). Samples were washed with tap water and transferred to 75% ethanol. Samples were made by cutting in the sagittal plane direction to include the intertransverse processes. The cut samples were embedded in wax and sectioned. The sections were cut sagittally from the block at 5 mm intervals with the microtome (LS-113; DAIWA-KOKI, Saitama, Japan). The sections were stained with hematoxylin-eosin, observed bone formation. Histological bone formation was assessed by three blinded independent observers. Histological fusion was defined as the percentage of new osteogenesis bridging formed between transverse processes and was scored on a five-point scale from 1 to 5 [35]. The scoring criteria were as follows: 1 = fibrocartilage tissue occupies less than 25% of the gap; 2 = fibula tissue occupies 25–75%; 3 = fibula tissue and bone tissue occupy 75–99%; 4 = bridged by newly osteogenesis tissue, but fusion mass is composed of the thin trabecular bone; 5 = completely bridged by sufficient mature bone tissue between transverse processes. The assessment was carried out at two locations in each sample, on the left and right sides.

### 4.10. Statistical Methods

The values in the study are presented as means with standard deviation. Statistical Package for the Social Sciences (SPSS) (V13; IBM Corporation, Armonk, NY, USA) was used for ANOVA analysis. In addition, Tukey’s honest significant difference (HSD) test was used as a post hoc analysis. Significant differences were defined as *p*-values less than 0.05. Kappa statistics were used as measures of interobserver reliability for three blinded, independent observers. Agreements were assessed as follows: poor, κ = 0–0.20; fair, κ = 0.21–0.40; moderate, κ = 0.41–0.60; substantial, κ = 0.61–0.80; and excellent, κ > 0.81. A value of one indicates perfect agreement between observers, while zero indicates that the numbers only matched by chance.

## Figures and Tables

**Figure 1 ijms-25-03655-f001:**
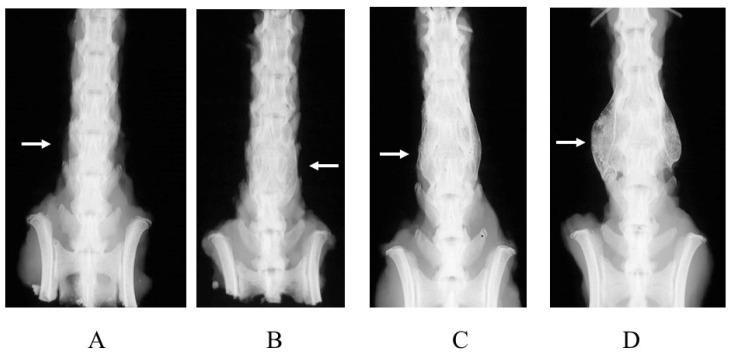
Radiographs of rat spines obtained after the rats were euthanized. (**A**) Group I (carrier alone), (**B**) Group II (3 μg BMP-2 without abaloparatide), (**C**) Group III (3 μg BMP-2 with abaloparatide) (3 times/week, total 120 μg/kg/week), and (**D**) Group IV (3 μg BMP-2 with abaloparatide) (6 times/week, total 120 μg/kg/week). White arrows indicate areas where surgery was performed and bone formation was expected.

**Figure 2 ijms-25-03655-f002:**
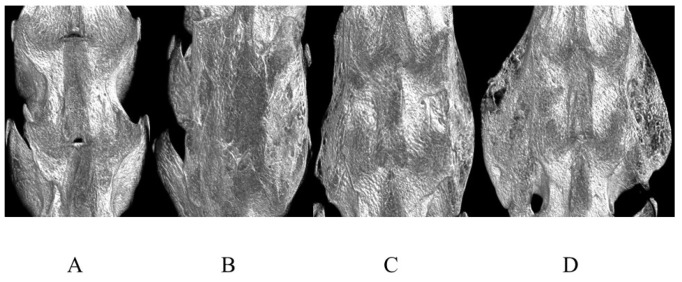
Representative 3D micro-CT anteroposterior images of rat spines. (**A**) Group I (carrier alone), (**B**) Group II (3 μg BMP-2 without abaloparatide), (**C**) Group III (3 μg BMP-2 with abaloparatide) (3 times/week, total 120 μg/kg/week), and (**D**) Group IV (3 μg BMP-2 with abaloparatide) (6 times/week, total 120 μg/kg/week).

**Figure 3 ijms-25-03655-f003:**
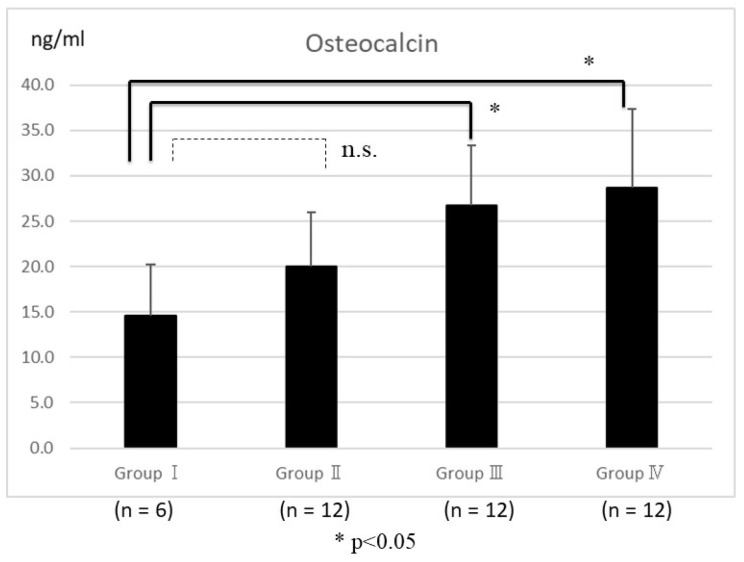
Enzyme-linked immunosorbent assay showed that serum osteocalcin levels were not significantly different in Group II and Group I. The significant differences were observed in Groups III and IV compared with Group I (*p* < 0.05). The bars represent the mean scores, and the error bars represent the standard deviation. The “n.s.” stands for not significant.

**Figure 4 ijms-25-03655-f004:**
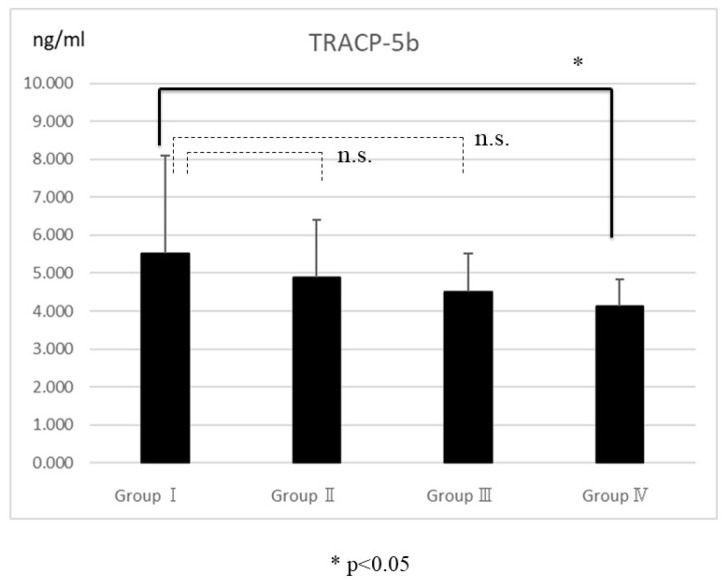
The serum levels of tartrate-resistant acid phosphatase-5b (TRACP5b) indicated that the differences between Groups I and IV were significant (*p* < 0.05). No significant differences were found between Groups I, II, and III. The bars represent the mean scores, and the error bars represent the standard deviations. The “n.s.” stands for not significant.

**Figure 5 ijms-25-03655-f005:**
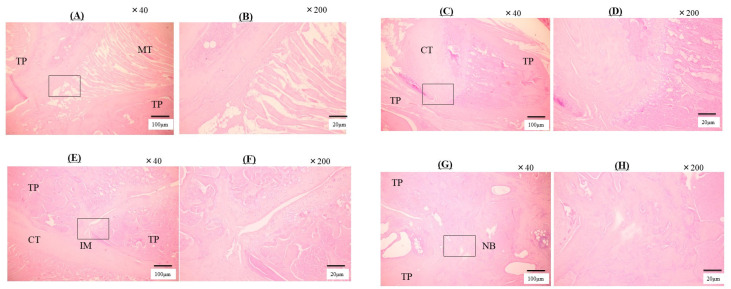
These are the tissue sections of the sagittal cut of the L4-L5 transverse process euthanized at 8 weeks post-surgery. (**A**) Group I, magnification ×40. (**B**) Group I, magnification ×200. These show the muscle tissue (MT) between the transverse processes (TP). A little new bone formation was observed, which may have resulted from the shedding of TP or from normal remodeling. (**C**) Group II, magnification ×40. (**D**) Group II, magnification ×200. The cartilaginous tissue (CT) is observed, but fibrosis tissue and muscle fiber filled in TP. (**E**) Group III, magnification ×40. (**F**) Group III, magnification ×200. Group III indicated the cartilaginous tissue and immature bone formation (IM) between TP. (**G**) Group IV, magnification ×40. (**H**) Group IV animals, magnification ×200. Group IV indicated that new bone (NB) formation bridging between TP, shows mature bone tissues and contracted bony trabeculae. The box shows the location magnified by a factor of 200.

**Table 1 ijms-25-03655-t001:** Spinal Fusion in Evaluations by Manual Palpation.

Group		No. Assessed	Assessed as Fused	Fusion Rate (%)
Group I	Carrier alone	12	0	0
Group II	3 μg BMP-2 without abaloparatide	24	4	16.7
Group III	3 μg BMP-2 with abaloparatide (3 times/week)	24	18	75.0 * (vs. Groups I and II)
Group IV	3 μg BMP-2 with abaloparatide (6 times/week)	24	16	66.7 * (vs. Groups I and II)

* *p* < 0.05. The fusion rate was significantly higher in Groups III and IV compared to Groups I and II in manual assessment.

**Table 2 ijms-25-03655-t002:** Radiographic scores at 8 weeks postoperatively.

Group		No. Assessed	Fusion Score (Mean ± SD)
Group I	Carrier alone	12	0.75 ± 1.21
Group II	3 μg BMP-2 without abaloparatide	24	1.44 ± 1.41
Group III	3 μg BMP-2 with abaloparatide (3 times/week)	24	3.12 ± 1.08 * (vs. Groups I and II)
Group IV	3 μg BMP-2 with abaloparatide (6 times/week)	24	3.04 ± 0.78 * (vs. Groups I and II)

* *p* < 0.05. The fusion scores were significantly higher in Groups III and IV compared to Groups I and II in radiographic assessment.

**Table 3 ijms-25-03655-t003:** Micro-CT-Based Histomorphometry.

Group		TV (mm^3^)	BV (mm^3^)	BV/TV (%)
Group I	Carrier alone	373.8 ± 75.4	58.7 ± 14.6	15.9 ± 4.16
Group II	3 μg BMP-2 without abaloparatide	398.7 ± 66.3	59.9 ± 8.39	15.2 ± 1.84
Group III	3 μg BMP-2 with abaloparatide (3 times/week)	582.2 ± 136.6 * (vs. Groups I, II, and IV)	87.5 ± 12.2 * (vs. Groups I and II)	16.1 ± 5.75
Group IV	3 μg BMP-2 with abaloparatide (6 times/week)	380.5 ± 78.6	86.7 ± 10.1 * (vs. Groups I and II)	23.2 ± 2.6 * (vs. Groups I, II, and III)

* *p* < 0.05. TV, tissue volume; BV, bone volume; BV/TV, bone volume fraction. TV in Group III was greater than that in other groups with significant differences. BV in Groups III and IV was greater than that in other groups with significant differences. BV/TV in Group IV was higher than in other groups with significant differences.

**Table 4 ijms-25-03655-t004:** Micro-CT-based histomorphometry of spines at 8 weeks.

Group		Tb. Th (mm)	Tb. N (1/mm)	Tb. Sp (mm)
Group I	Carrier alone	0.26 ± 0.01	0.62 ± 0.15	1.77 ± 0.43
Group II	3 μg BMP-2 without abaloparatide	0.21 ± 0.05	0.75 ± 0.11	1.54 ± 0.12
Group III	3 μg BMP-2 with abaloparatide (3 times/week)	0.20 ± 0.04	0.77 ± 0.16	1.20 ± 0.17 * (vs. Group I)
Group IV	3 μg BMP-2 with abaloparatide (6 times/week)	0.25 ± 0.04	0.96 ± 0.20 * (vs. Group I)	1.13 ± 0.19 * (vs. Group I)

* *p* < 0.05. Tb. Th, trabecular thickness; Tb. N, trabecular number; Tb. Sp, trabecular spacing. Tb. N in Group IV was significantly higher than that in Group I. Tb. Sp in Groups Ⅲ and IV was of significantly smaller values than those in Group I.

**Table 5 ijms-25-03655-t005:** Histological fusion score at 8 weeks.

Group		Fusion Scores (Mean ± SD)
Group I	Carrier alone	1.40 ± 0.69
Group II	3 μg BMP-2 without abaloparatide	1.91 ± 1.16
Group III	3 μg BMP-2 with abaloparatide (3 times/week)	3.91 ± 1.16 * (vs. Groups I and II)
Group IV	3 μg BMP-2 with abaloparatide (6 times/week)	4.16 ± 1.03 * (vs. Groups I and II)

* *p* < 0.05. The fusion scores were significantly higher in Groups III and IV compared to Groups I and II in histological assessment.

## Data Availability

The data are contained within this article.

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
