# Peer review of "Optimal Intermittent Administration Interval of Abaloparatide for Bone Morphogenetic Protein-Induced Bone Formation in a Rat Spinal Fusion Model"

_ijms, 2024, doi:10.3390/ijms25073655_

Round 1

Reviewer 1 Report

Comments and Suggestions for Authors

The study aims to determine the optimal dosing interval for the combination of rhBMP-2 and abaloparatide in a rat spinal fusion model. Even if the dose and interval are optimized, these parameters are definitely different from human.

The study design failed to address this research question, since group 3 and 4 are different in both dose and interval. Thus, the study design cannot address exactly which factor is responsible for the observed difference between group 3 and 4.

The study failed in properly evaluate spinal fusion. Particularly, the 'manual assessment of movement' is completely biased and imprecise. A proper mechanical test is required. The HE staining is not the best method to evaluate bone formation.

The method of surgery is completely missing. The timeline of drug administration is not described in details. The animal age and sex should be considered since they will interfere the results.

Comments on the Quality of English Language

The scientific description needs higher clarity. Examples include but not limited to: in line 88 'spine was scarred', in line 106 'quality of the spinal fusion area'. What is scar defined? And which quality was evaluated. 

Reviewer 2 Report

Comments and Suggestions for Authors

Abstract is not clearly written. There are some very confusing sentences like "Group I was implanted with the implants". 

Otherwise, article is of high quality. Materials and methods used in this study are appropriate for assessment of therapy. Results of this study are of certain value for the field. BMP2 osteogenic potential is clear, the same is for abaloparotide. Therefore, this result is somehow expected. Findings should be discussed more in details, for example compared to recent studies describing potential of BMP6 in spinal fusion procedures.

Round 2

Reviewer 1 Report

Comments and Suggestions for Authors

The answers to the concerns are mostly referring to other literatures. Not sure if this is normal that a proper understanding of this paper need a systematic review of other literatures.

Would like to leave the editor to decide whether to accept the paper for publication.

Comments on the Quality of English Language

NA

Author Response

IJMS (ISSN 1422-0067) ijms-2864164 Title: Optimal Intermittent Administration Interval of Abaloparatide for Bone Morphogenetic Protein-Induced Bone Formation in a Rat Spinal Fusion Model   We have addressed each of the points raised by the reviewers and would like to submit a revised manuscript for publication in International Journal of Molecular Sciences. The reviewers' comments and the corresponding changes in the revised manuscript are detailed below:   Dear Reviewer 1: Thank you for your comments regarding our paper. In response to your comments, I added the following sentence at the paragraph.   Page 8, “Finally, although the total doses were aligned in this study, it remains debatable whether more frequent dosing is better, or the per-dose doses were more appropriate in bone formation. Bone formation was more enhanced in Group â…£, which the frequency of dosing was increased, than in Group â…¢, which single doses were increased. So, the frequency of administration was determined to have a greater impact in this study. Additional studies under various conditions are needed in the future.”   Once again, thank you for your consideration of our paper. We hope the reviewers’ concerns are addressed in the revised manuscript and our answers are satisfactory.  Sincerely yours,